# The Risk Factors Associated with the Carriage to Critical Antimicrobial-Resistant *Escherichia coli* in Healthy Household Dogs: A One Health Perspective

**DOI:** 10.3390/ani15101357

**Published:** 2025-05-08

**Authors:** Carlos Alejandro Zelaya, Gabriel Arriagada, Rosario Medina, Beatriz Escobar, Fernando Sánchez, Nicolás Galarce, Lisette Lapierre

**Affiliations:** 1Programa de Doctorado en Bioinformática y Biología de Sistemas, Universidad Andrés Bello, Santiago 7510000, Chile; c.zelayamenjivar@uandresbello.edu; 2Departamento de Medicina Preventiva Animal, Facultad de Ciencias Veterinarias y Pecuarias, Universidad de Chile, Santiago 8820000, Chile; rosario.medina@ug.uchile.cl (R.M.); beatrizescobar@uchile.cl (B.E.); fernando.sanchez@ug.uchile.cl (F.S.); ngalarce@uchile.cl (N.G.); 3Instituto de Ciencias Agroalimentarias, Animales y Ambientales, Universidad de O’Higgins, San Fernando 3071991, Chile

**Keywords:** dogs, *Escherichia coli*, antibiotic resistance, risk factors

## Abstract

*Escherichia coli* is a bacterium commonly found in the intestinal microbiota of humans and animals; however, certain isolates can act as opportunistic pathogens. Some of these isolates are resistant to critically important antimicrobials, complicating the treatment of infections. In this study, we investigated factors associated with the carriage of antimicrobial-resistant *E. coli* in healthy household dogs in Chile. We included dogs that had not received any medications for several weeks, analyzed fecal samples, and collected information from owners regarding the dogs’ origins and living conditions. The key findings revealed that dogs purchased from pet stores or kennels had higher rates of resistance compared to adopted dogs. Other relevant factors included previous hospital visits, age, and sex. Interestingly, living with a healthcare worker or having occasional contact with free-roaming animals appeared to be associated with a lower risk of carrying resistant bacteria. These results suggest that responsible pet acquisition and prudent antimicrobial use can help reduce the spread of resistant bacteria. This study reinforces the importance of addressing antimicrobial resistance from a One Health perspective, recognizing the interconnectedness of human, animal, and environmental health.

## 1. Introduction

Currently, dogs maintain close contact with their owners, facilitating the mutual exchange of microorganisms, including antibiotic-resistant bacteria (ARB). These ARB can be transmitted between interdependent hosts and further disseminated into the environment, contributing to the global rise in antimicrobial resistance (AMR) [1]. Among the bacteria exhibiting resistance and multidrug resistance (MDR) is *Escherichia coli*, a common component of the normal microbiota in various animal species, including humans and dogs. This bacterium has the capacity to acquire new antimicrobial resistance mechanisms from other bacteria and disseminate them horizontally [2].

According to the World Health Organization (WHO), AMR and the proliferation of resistant bacteria represent some of the most critical threats to global public health. In response, since 2014, surveillance and control programs targeting AMR in commensal and zoonotic bacteria from production animals have been implemented. However, companion animals are generally excluded from these programs, resulting in a lack of official data on AMR in this population [3,4]

Surveillance of AMR in owned dogs is essential, as resistant bacteria in pets may transfer resistance genes to human-associated bacteria and vice versa, potentially complicating the treatment of infections in both hosts. Additionally, the rise in AMR among bacteria isolated from dogs poses a significant threat to their health by limiting available therapeutic options [1]. Monitoring AMR in isolates from companion animals, along with identifying factors influencing its occurrence, is crucial for both public and animal health. This information supports veterinarians in making safer and more targeted antimicrobial prescriptions. Without such data, clinicians often resort to using the newest broad-spectrum antimicrobials, which can exacerbate AMR issues over time [5,6]. The WHO and the World Organization for Animal Health (WOAH) emphasize the protection of critically important antimicrobials, classified based on their necessity for treating severe infections in both veterinary and human medicine when no effective alternatives are available [7,8]. Therefore, the responsible use of critical antibiotics in veterinary medicine is a priority under the One Health framework.

With the significant rise in AMR and the limited success of current strategies to prevent and reduce it, identifying risk factors associated with the presence of ARB in pets is essential. Understanding these factors can help veterinarians and pet owners implement targeted interventions to mitigate AMR effectively. A One Health-centered approach to AMR control should consider potential risk factors for its spread in animals, including intrinsic animal characteristics, clinical conditions, environmental exposures, and owner-related habits. Consequently, we designed this study to identify actionable risk factors linked to resistant *E. coli* in healthy household dogs, providing evidence for targeted AMR control strategies in companion animals.

## 2. Materials and Methods

### 2.1. Study Design

This study followed a cross-sectional design. Dogs were selected from veterinary clinics located in the Metropolitan Region (MR), the capital region of Chile. The sample size was calculated to include 386 dogs, using a formula for estimating proportions as described by Dohoo et al. [9]. This calculation assumed no prior knowledge of the proportion of dogs carrying antibiotic-resistant *E. coli* (a priori proportion set at 50%), with a precision of 5% and a 95% confidence interval.

Dogs were selected through a multi-stage sampling process, with veterinary clinics serving as the primary sampling units and dogs as the secondary units. The sampling frame for veterinary clinics was constructed using all businesses labeled as “clínica veterinaria” on Google Maps. For each clinic, the municipality and macro-area were recorded. Using a stratified random sampling approach, four to five veterinary clinics were selected from each of the seven macro-areas of the MR. Directors or owners of the selected clinics were contacted and invited to participate in the study. If a clinic did not respond or declined, another clinic from the same macro-area was randomly selected until the target sample size was achieved. Within each participating clinic, 10 to 12 dogs were selected in order of arrival, provided they met the following criteria: (1) being healthy upon clinical examination, (2) having no history of antibiotic treatment in the previous four weeks, and (3) residing in the same macro-area as the clinic.

### 2.2. Sample Collection

Dogs attending veterinary clinics during 2021–2022 were sampled via rectal swabbing, with prior institutional approval (permit code 21439-VET-UCH) and written informed consent from the owners. Rectal swabs with Cary Blair transport medium (Copan^®^, Murrieta, CA, USA) were collected from clinically healthy dogs of any age and sex. The swab was inserted approximately 2 cm into the rectum and gently rotated for 10 s. All samples were immediately refrigerated and transported to the laboratory within four h of collection.

### 2.3. Isolation and Identification of E. coli

Swabs containing dog feces were enriched in 9 mL of tryptone soy broth (Becton, Dickinson & Co., Franklin Lakes, NJ, USA), homogenized, and incubated overnight at 42 °C. Following enrichment, 50 μL of each culture was plated onto four MacConkey agar plates (Becton, Dickinson & Co., Franklin Lakes, NJ, USA) and incubated at 37 °C for 18–24 h to facilitate the detection of *E. coli*. Three of the plates were supplemented with different antibiotics: amoxicillin-clavulanic acid (AMC, 16 μg/mL, Acros Organics^®^, Geel, Belgium), ceftazidime (CTZ, 2 μg/mL, Sigma^®^, St. Louis, MO, USA), and enrofloxacin (ENR, 2 μg/mL, Sigma^®^). One plate without antibiotics served as a control to isolate antibiotic-susceptible strains and verify growth.

These antibiotics were selected due to their high importance in human medicine and critical importance in animal health, as well as their frequent use in companion animal clinical practice in Chile. According to the WHO, aminopenicillins combined with beta-lactamase inhibitors are classified as highly important, whereas the WOAH categorizes them as critically important for animal health [5,7,8]. Antibiotic concentrations incorporated into the selective plates were chosen based on established clinical breakpoints to ensure appropriate selection pressure for the detection of resistant isolates [10].

When bacterial growth was observed, one colony exhibiting typical *E. coli* morphology was selected from each antibiotic-supplemented selective agar plate for further analysis. The identity of the selected colonies was confirmed as *E. coli* by PCR. DNA from all presumptive colonies was extracted using the Wizard^®^ Genomic DNA Purification Kit (Promega, Madison, WI, USA), following the manufacturer’s instructions. DNA quality and concentration (260/280 absorbance ratio) were measured using a nanodrop (NANO-400 micro-spectrophotometer, Hangzhou Allsheng Instruments Co., Hangzhou, China). Samples with an absorbance ratio within the optimal range (1.8–2.0) were stored at –20 °C for molecular analysis.

PCR was carried out using a LifeECO^®^ thermal cycler (Hangzhou Allsheng Instruments Co.) according to the protocol described by Chen and Griffiths [11]. Table 1 details the primers used and the expected amplicon sizes. The *E. coli* ATCC 25922 strain served as the positive control, while *Salmonella typhimurium* ATCC 14028 was used as the negative control [12].

### 2.4. Phenotypic Antimicrobial Resistance Characterization

All colonies that grew on plates supplemented with AMC, CTZ, and ENR and were confirmed as *E. coli* were analyzed for AMR using the automated VITEK2 system (bioMérieux, Marcy-l’Étoile, France). The minimal inhibitory concentration (MIC) of the antimicrobials was determined for each isolate using the AST-GN98 card, following the manufacturer’s instructions and applying clinical breakpoint values as outlined in the guidelines of the Clinical and Laboratory Standards Institute (CLSI) [10]. Based on the MIC values obtained for each antimicrobial, isolates were classified as susceptible, intermediate, or resistant. For the purposes of this study, isolates categorized as intermediate were considered resistant [13].

### 2.5. Risk Factors

Risk factors associated with AMR in dogs were evaluated using a structured questionnaire divided into four thematic modules: Module 1, “General Information About Caretakers”; Module 2, “General Information About the Pet”; Module 3, “Clinical Information About the Pet”; and Module 4, “Environmental Information Related to the Pet”. Data collection was conducted in person during sample collection, via email, or through telephone interviews with the caretakers of the selected dogs.

To ensure data quality, complete responses were mandatory for all study participants. Any incomplete responses resulted in the exclusion of the respective individuals from the risk analysis. All collected data were securely stored in an Excel database for subsequent analysis and management.

### 2.6. Statistical Analysis

To evaluate potential risk factors associated with the presence of antibiotic-resistant *E. coli* in household dogs, three multivariable logistic regression models were constructed. Each model assessed the relationship between the log-odds of a dog being a carrier of *E. coli* resistant to AMC, ENR, or CTZ and a set of potential risk factors. The analysis followed a four-stage approach:(1)The initial association between each potential risk factor and the likelihood of *E. coli* resistance to each antibiotic was evaluated using a chi-square test. Risk factors with a *p*-value lower than 0.25 were included in the multivariable logistic regression (MLR) analysis.(2)Final models were developed using a stepwise backward elimination method, with adjustments for potential confounding variables.(3)Collinearity in the final models was evaluated using Pearson’s correlation test, examining correlation values between predictors.(4)Model fit was assessed using Pearson’s chi-square test. The strength of the association between risk factors and the presence of resistant *E. coli* was represented by the final model estimates, expressed as odds ratios (ORs). Statistical significance was determined using a *p*-value threshold of <0.05. All statistical analyses were conducted using RStudio software (Version 2023.09.1).

## 3. Results

### 3.1. Characterization of Sampled Dogs

Of the 386 dogs enrolled, 263 (68.1%) were included in the final analysis due to complete survey responses. The sampled dogs were from households located in various settings within the MR of Chile, encompassing both urban and rural areas. The age distribution included 53% adults (1–7 years old), 20% puppies (less than 1 year old), and 27% seniors (older than 7 years). Most dogs were mixed breed (60%), while 40% were purebred. Female dogs accounted for 55% of the sampled population.

Regarding diet, most dogs were primarily fed commercial pellets (85%), while 15% received a mixed diet of commercial and homemade food. Additionally, 20% of owners reported including raw meat or bones in their dogs’ diets. Although some dogs had contact with other animals outside of the owner’s supervision, most were walked on a leash, and only a minority of owners reported observing their dogs drinking from outdoor water sources. Further details on the characteristics of the sampled dog population are provided in Appendix A.

Of the 301 *E. coli* isolates analyzed, 69 (22.9%) were resistant to one of the three antimicrobials incorporated into the selective media, 16 isolates (5.3%) showed resistance to two antimicrobials, and 4 isolates (1.3%) were resistant to all three. In particular, the detection rates of AMC-, CTZ-, and ENR-resistant isolates were 13.3% (95% CI: 9.45–17.12), 5.9% (95% CI: 3.30–8.66), and 18.3% (95% CI: 13.91–22.64), respectively (Figure 1). The corresponding MIC values for each of these antimicrobials are shown in Table 2, along with their CLSI interpretive breakpoints.

Statistical analysis was conducted using data from the 263 complete questionnaires (Appendix A). Variables with more than two response options were categorized into binary or three-level variables based on biological criteria, ensuring sufficient statistical power to detect significant effects when analyzing multiple categorical variables simultaneously.

### 3.2. Risk Factors Associated with AMC-Resistant E. coli Isolates

Of all of the risk factors assessed in the questionnaire, only five were included in the MLR analysis. These factors were as follows: (1) whether a household member works in a human or veterinary healthcare facility, (2) the type of diet the dog receives, (3) the inclusion of raw meat or bones in the diet, (4) contact with animals without the owner’s supervision, and (5) whether the dog is walked on a leash (Table 3).

The MLR analysis identified several risk factors associated with AMC-resistant *E. coli* in dogs (Table 4). Dogs from households where owners or household members worked in human or veterinary healthcare facilities (X6-Yes: OR = 0.32; 95% CI: 0.13–0.72) had significantly lower odds of carrying AMC-resistant *E. coli* (*p* = 0.01). Furthermore, dogs whose diets included raw meat or bones (X28-Yes: OR = 0.10; 95% CI: 0.01–0.56) were also significantly less likely to carry AMC-resistant *E. coli* (*p* = 0.034). Similarly, dogs that had contact with animals outside of the owner’s supervision (X35-Yes: OR = 0.32; 95% CI: 0.13–0.74) showed a significantly reduced likelihood of carrying AMC-resistant *E. coli* (*p* = 0.01). In addition, dogs that were consistently walked on a leash (X38-Yes: OR = 0.26; 95% CI: 0.11–0.60) demonstrated significantly lower odds of carrying these resistant *E. coli* strains (*p* = 0.002).

On the other hand, the primary diet type was not significantly associated with the carriage of AMC-resistant *E. coli*. This was observed in dogs primarily fed commercial food (X25-Commercial food: OR = 0.42; 95% CI: 0.12–1.53; *p* = 0.16) or homemade food (X25-Homemade food: OR = 194,866,108.8; 95% CI: 2.92 × 10^−17^–1.01 × 10^249^; *p* = 0.99).

The model’s goodness of fit was assessed using Pearson’s chi-square test, which produced a chi-square statistic of 217.30 with 293 degrees of freedom and a *p*-value of 0.99, indicating an excellent fit (Appendix A).

### 3.3. Risk Factors Associated with CTZ-Resistant E. coli Isolates

Of all of the risk factors assessed in the questionnaire, only five were included in the MLR analysis. These factors were as follows: (1) whether a household member works in a human or veterinary healthcare facility, (2) the sex and age of the dog, (3) whether the dog was purchased or adopted, (4) whether the dog lives with other pets that have been hospitalized in the past year, and (5) whether the dog is taken for walks during the week (Table 5).

The MLR analysis identified several risk factors associated with the carriage of CTZ-resistant *E. coli* in dogs (Table 6). The sex of the dog was identified as a potential risk factor, with male dogs (X10-Male: OR = 3.07; 95% CI: 1.02–10.29) showing a higher likelihood of carrying CTZ-resistant *E. coli* (*p* = 0.05), although this association was marginally significant. Additionally, the age of the dog played a significant role, as dogs under two years old (X11-Under 2 years old: OR = 3.88; 95% CI: 1.26–12.86) had significantly higher odds of carrying CTZ-resistant *E. coli* (*p* = 0.02), suggesting that younger dogs may be more susceptible to colonization by these resistant isolates. The origin of the dog also demonstrated a strong association, with purchased dogs (X14-Purchase: OR = 6.04; 95% CI: 1.88–21.32) exhibiting significantly increased odds of carrying CTZ-resistant isolates (*p* = 0.003).

On the other hand, household factors did not show statistically significant associations with the carriage of CTZ-resistant *E. coli*. Dogs living with other pets that had been hospitalized in the past year (X36-Yes: OR = 7.30 × 10^−9^; *p* = 0.99) did not exhibit a significant relationship with the carriage of CTZ-resistant isolates. Similarly, dogs that regularly went for walks (X37-Yes: OR = 88,661,041.94; 95% CI: 3.28 × 10^−65^–2.81 × 10^284^; *p* = 0.99) showed extremely large odds ratios, but these results lacked statistical significance, likely due to collinearity or low representation within the dataset.

The model’s goodness of fit was assessed using Pearson’s chi-square test, resulting in a chi-square statistic of 130.37 with 292 degrees of freedom and a *p*-value of 1, indicating an excellent fit (Appendix A).

### 3.4. Risk Factors Associated with ENR-Resistant E. coli Isolates

Of all of the risk factors assessed in the questionnaire, only six were included in the multivariable logistic regression (MLR) analysis. These factors were as follows: (1) whether a household member works in a human or veterinary healthcare facility, (2) the size of the dog, (3) the length of time the owner has been living with the dog, (4) whether the dog has ever been hospitalized, (5) the primary location where the owner purchases pet food, and (6) whether the dog receives treats (Table 7).

The MLR analysis identified several risk factors associated with ENR-resistant *E. coli* in dogs (Table 8). Dogs whose owners or household members worked in human or veterinary healthcare facilities (X6-Yes: OR = 0.17; 95% CI: 0.07–0.37) had significantly lower odds of carrying ENR-resistant *E. coli* (*p* = 0.00002), indicating a potential protective effect. Similarly, dogs that had been living with their owners for 5–10 years (X15-Between 5 and 10 years: OR = 0.27; 95% CI: 0.11–0.65) or for more than 10 years (X15-More than 10 years: OR = 0.27; 95% CI: 0.07–0.88) exhibited significantly reduced odds of carrying ENR-resistant *E. coli* (*p* = 0.005 and *p* = 0.041, respectively).

The source of commercial pet food also appeared to influence the carriage of ENR-resistant *E. coli*. Dogs whose owners purchased pet food from mass sales commercial establishments (X26-Mass sales: OR = 0.14; 95% CI: 0.03–0.50) had significantly lower odds of carrying ENR-resistant *E. coli* (*p* = 0.004). However, purchasing pet food from small local stores or fairs (X26-Small-sized commercial establishment: OR = 0.36; 95% CI: 0.06–1.90) did not show a statistically significant association (*p* = 0.235).

Conversely, dogs that had been hospitalized at least once in their lifetime (X17-Yes: OR = 4.24; 95% CI: 1.91–9.88) exhibited significantly higher odds of carrying ENR-resistant *E. coli* (*p* = 0.0005). Additionally, providing treats to the dog (X33-Yes: OR = 2.94; 95% CI: 1.10–8.85) was associated with significantly increased odds of carrying ENR-resistant *E. coli* (*p* = 0.041), suggesting that diet-related factors may contribute to colonization by ENR-resistant isolates. In contrast, the size of the dog (X12-Medium: *p* = 0.264; X12-Small: *p* = 0.464) was not significantly associated with the carriage of ENR-resistant *E. coli*.

The model’s goodness of fit was evaluated using Pearson’s chi-square test, which yielded a chi-square statistic of 188.27 with 290 degrees of freedom and a *p*-value of 1.0 (Appendix A).

## 4. Discussion

Understanding the risk factors associated with the transmission of ARB in dogs is essential for elucidating their epidemiology and informing effective control strategies. This study investigated a range of intrinsic, clinical, and environmental factors influencing the carriage of *E. coli* resistant to three antimicrobials regarded as critically important for animal and human health. Our findings highlight key determinants contributing to the carriage of resistant *E. coli* in companion animals, providing valuable insights for targeted interventions.

A systematic review by Karalliu et al. [14], analyzing 40 published studies, identified the most frequently investigated risk factors for the transmission of antibiotic-resistant *Enterobacterales* in dogs. These included antimicrobial use (28 studies), age (24), sex (22), hospitalization (19), and feeding with raw diets (14). Among these, antimicrobial use was the most commonly and significantly associated risk factor (19 out of 28 studies), followed by raw diet consumption (9 out of 14) and prior hospitalization (8 out of 19).

In our study, we examined a broad range of risk factors by surveying the owners of healthy dogs. We isolated *E. coli* from the dogs’ feces and assessed its susceptibility to three critically important antimicrobials. The risk factors identified by Karalliu et al. [14] were included; however, in our study, only prior hospitalization and sex were significantly associated with the carriage of resistant *E. coli*. It is important to note that regarding prior hospitalization, the questionnaire did not inquire about the duration or the underlying cause of the hospitalization.

### 4.1. Risk Factors Associated with ENR Resistance

Our study identified prior hospitalization as a significant risk factor for the carriage of ENR-resistant *E. coli*, consistent with previous research. For instance, Gibson et al. [15] conducted a case–control study involving 90 cases and 93 controls, evaluating risk factors in dogs admitted to a university veterinary hospital. They found that hospitalization for more than six days, prior treatment with cephalosporins, and treatment with cephalosporins or metronidazole during hospitalization were associated with an increased risk of rectal carriage of MDR *E. coli*. Similarly, Dazio et al. [16] assessed the prevalence of ARB carriage in dogs and cats admitted to veterinary clinics. They reported that 15.5% of companion animals carried MDR bacteria upon admission, which increased to 32.1% at discharge, with an acquisition rate of 28.3%. The most frequently acquired hospital-associated pathogens were extended-spectrum β-lactamase-producing *E. coli* (ESBL-*E. coli*; 17.3%) and β-lactamase-producing *Klebsiella pneumoniae* (13.7%). These findings emphasize the importance of effective patient management and prudent antibiotic use during hospitalization to prevent the emergence of nosocomial infections in veterinary hospitals.

Conversely, prolonged pet ownership was identified as a protective factor in this study. Dogs that had lived with their owners for more than five years, particularly those for over ten years, were less likely to carry ENR-resistant *E. coli*. This may be due to responsible pet ownership, where long-term caregivers implement better preventive measures, thereby reducing their dogs’ exposure to risk factors that facilitate the acquisition of ARB.

Another risk factor was the source of the dogs’ food. Dogs fed non-raw diets, such as kibble purchased from large-scale commercial establishments, were less likely to carry ENR-resistant *E. coli* compared to those fed through other sources. In Chile, dry kibble is the most widely consumed commercial dog food, and its production process, particularly heat treatment, likely reduces bacterial contamination [17]. Additionally, commercial kibble undergoes rigorous quality control measures to ensure safety, further minimizing the risk of contamination with resistant bacteria [18]. A study on the carriage of antibiotic-resistant *E. coli* in dogs also found that commercial dry diets serve as a protective factor [19]. In contrast, food sold in smaller retail establishments is often available in bulk and may lack proper microbiological quality control, increasing the likelihood of contamination with resistant bacteria.

### 4.2. Risk Factors Associated with AMC Resistance

Our study found that occasional contact between sampled dogs and unsupervised animals outside their household was identified as a protective factor for significantly reduced carriage of AMC-resistant *E. coli*. This may be explained by exposure to environmental bacteria or a more diverse microbiota, which could promote bacterial competition and limit the dissemination of AMC-resistant isolates. For instance, the presence of a complex microbial community derived from pig feces was shown to reduce the selection of *E. coli* mutants resistant to kanamycin or gentamicin through two key mechanisms. First, the increased tolerance of *E. coli* embedded in this mixed-species consortium diminished the relative fitness advantage of resistant mutants over their susceptible counterparts, thereby weakening the selection pressure for resistance and shifting the emergence of resistant mutants to higher antibiotic concentrations. Second, intensified competition for resources amplified the fitness cost of resistance mechanisms, further limiting the advantage of resistant isolates under subinhibitory antibiotic concentrations [20]. Similarly, reduced nutrient availability has been shown to increase the fitness cost of rifampicin resistance, emphasizing the role of ecological pressures in modulating bacterial adaptation [21].

The dog population in our study primarily consisted of urban dogs. Therefore, in this context, interactions with unsupervised animals mainly involved other urban dogs, cats, or birds that roam freely within the community or belong to different households, rather than wildlife, farm animals, or livestock.

Regarding the dogs’ living environment, Sealey et al. [22] compared the carriage of third-generation cephalosporin-resistant *E. coli* between dogs from urban and rural settings. In rural dogs, the presence of cephalosporin-resistant *E. coli* harboring *blaCTX-M* was phylogenetically associated with interactions between dogs and local livestock and epidemiologically linked to raw meat feeding. In contrast, urban dogs exhibited more varied risk sources, including environmental exposures such as rivers. Similarly, Caxito et al. [23] analyzed extended-spectrum cephalosporin-resistant *E. coli* isolates obtained from rectal swabs of dogs in southeastern Brazil. Their findings showed that dogs with livestock contact were eight times more likely to carry these resistant isolates compared to those without such exposure.

### 4.3. Risk Factors Associated with CTZ Resistance

Several factors were associated with an increased likelihood of dogs carrying *E. coli* resistant to CTZ. Male dogs showed significantly higher carriage rates of CTZ-resistant *E. coli*. This finding aligns with Umeda et al. [24], who reported a higher prevalence of cephalosporin-resistant *Enterobacteriaceae* in male dogs. However, contrasting findings by Belas et al. [25] suggest that female dogs exhibit a higher risk of carrying *E. coli* producing plasmid-mediated AmpC, potentially due to their increased susceptibility to urinary tract infections.

Age also played a critical role in CTZ resistance. Dogs younger than two years old had a significantly higher probability of carrying resistant *E. coli* compared to those between two and ten years old. This may be attributed to younger dogs’ higher activity levels, increased exposure to environmental bacteria, and a greater likelihood of sustaining wounds that require antibiotic treatment. While our study found higher resistance in younger dogs, Ekakoro et al. [26] and Groat et al. [27] reported greater AMR in *E. coli* isolates from geriatric dogs (over ten years old), attributing it to immunosenescence and cumulative antibiotic exposure over their lifetime. This age-dependent pattern suggests that different resistance mechanisms may dominate at various life stages, with young dogs facing exposure-driven resistance, while older dogs accumulate resistance through prolonged antibiotic selection pressure.

### 4.4. Protective Factors: Healthcare Workers and Diet

Our study identified cohabitation with a person working in a healthcare setting, whether veterinary or human, as a protective factor for significantly reduced carriage of *E. coli* resistant to ENR and AMC. This may be explained by the increased awareness among individuals with training, education, or experience in the health field regarding the adverse effects of inappropriate antimicrobial use. Such awareness could contribute to reduced exposure of dogs to unnecessary antibiotic treatments, lower antibiotic usage, or improved hygiene practices at home.

Additionally, dogs fed mixed diets, combining kibble with raw food, had a lower probability of carrying *E. coli* resistant to AMC compared to those exclusively fed raw diets. This aligns with multiple studies highlighting the risks of raw feeding. Among these, Groat et al. [27] analyzed rectal swabs from dogs exclusively fed raw diets (n = 114) and those that were not (n = 76). Their results indicated that dogs consuming raw diets were more likely to harbor antimicrobial-resistant *E. coli*, resistant to third-generation cephalosporins or MDR isolates, compared to those on non-raw diets. Similarly, Morgan et al. [28] in the UK found that dogs fed raw diets had significantly higher fecal carriage of antimicrobial-resistant *E. coli* (*p* < 0.001), including ESBL-producing isolates, compared to dogs consuming non-raw diets. Additionally, Van den Bunt et al. [29] demonstrated that feeding processed dry food reduced the likelihood of detecting β-lactamase-producing *E. coli* in companion dogs in the Netherlands. Collectively, these studies highlight the significant impact of diet on the carriage of ARB in dogs, emphasizing the potential risks associated with raw feeding practices.

### 4.5. Public Health and Veterinary Implications

Our findings highlight significant zoonotic transmission risks associated with ARB in dogs. A notable 2017 outbreak in the United States illustrates this concerning dynamic. The Florida Department of Health reported an outbreak of antibiotic-resistant *Campylobacter jejuni* infections in humans to the CDC. The investigation traced the outbreak to a chain of pet stores, where dogs sold in the stores carried the same antibiotic-resistant isolates found in infected humans, including store employees and customers who had purchased the dogs. Additionally, a review of store records revealed that out of 149 puppies investigated, 142 (95%) had received one or more antibiotic treatments before being sold [30]. This case exemplifies the critical One Health connection between veterinary practices and human public health, emphasizing the need for coordinated surveillance and prevention strategies.

## 5. Conclusions

This study identified key risk and protective factors influencing the carriage of *E. coli* resistant to critically important antibiotics in healthy household dogs. Hospitalization emerged as a major risk factor, while living with health-trained individuals, occasional contact with free-roaming animals, and certain diets appeared to have protective effects.

These results highlight the complexity of AMR transmission in pets and the value of adopting a One Health approach. Educating owners—particularly regarding raw feeding and breeding practices—may help reduce AMR. Despite the limited sample size, the findings provide a solid foundation for guiding policies and future research aimed at controlling AMR in companion animals.

Although the multivariable models demonstrate good overall fit, as indicated by residual deviance-to-degrees of freedom ratios close to or below 1 and highly significant likelihood ratio tests, caution should still be exercised when interpreting the results. The relatively small sample size and the regional specificity of the study population, limited to central Chile, may affect the generalizability of the findings. Moreover, cultural practices, patterns of antimicrobial usage, and veterinary regulations vary across countries and could influence risk factor profiles. Additionally, the presence of potential unmeasured confounders cannot be entirely excluded.

## Figures and Tables

**Figure 1 animals-15-01357-f001:**
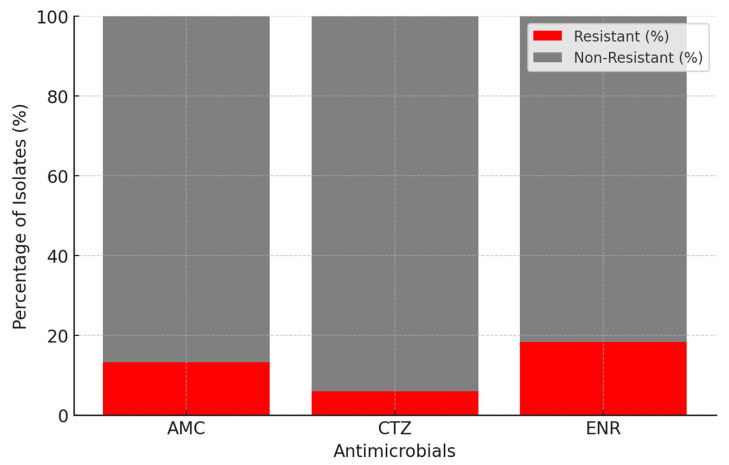
A stacked bar chart showing antimicrobial resistance detection in *E. coli* isolates (n = 301). The chart displays the number of resistant and non-resistant isolates for each antimicrobial tested: amoxicillin-clavulanic acid (AMC), ceftazidime (CTZ), and enrofloxacin (ENR).

**Table 1 animals-15-01357-t001:** The primers used for the identification of *E. coli* by PCR. The table lists the sequences of the forward (F) and reverse (R) primers, the expected amplicon sizes in base pairs (bp), and the corresponding references.

Gene	Primers	Expected Product Size (bp)	Reference
*uspA*	F: CCGATACGCTGCCAATCAGT R: ACGCAGACCGTAGGCCAGAT	884	[11]
*uidA*	F: TATGGAATTTCGCCGATTTT R: TGTTTGCCTCCCTGCTGCGG	166	[11]

**Table 2 animals-15-01357-t002:** Distribution of minimum inhibitory concentration (MIC) values among *E. coli* isolates for each antibiotic tested. Breakpoint values were interpreted according to CLSI VET01S guidelines when available.

Antibiotic/Breakpoints	MIC µg/mL	Number of Isolates	Interpretation (S/I/R)
Amoxicillin-clavulanic acid (AMC)/ S ≤ 8 µg/mL I = 16 µg/mL R ≥ 32 µg/mL	0.25	228	S
	2	28	S
	4	5	S
	16	21	I
	32	19	R
Ceftazidime (CTZ)/ S ≤ 4 µg/mL I = 8 µg/mL R ≥ 16 µg/mL	0.25	269	S
	0.5	6	S
	1	18	S
	2	5	S
	4	3	S
	8	12	I
	32	6	R
Enrofloxacin (ENR)/ S ≤ 0.5 µg/mL I = 1 µg/mL R ≥ 2 µg/mL	0.25	233	S
	0.5	13	S
	1	31	I
	2	1	R
	4	23	R

Abbreviations: MIC, minimum inhibitory concentration; S, susceptible; I, intermediate; R, resistant.

**Table 3 animals-15-01357-t003:** The results of the likelihood ratio test (LRT) for variables included in the multivariable logistic regression model for AMC resistance *E. coli* in household dogs. The table presents the degrees of freedom (Df), deviance, Akaike Information Criterion (AIC), LRT values, and *p*-values (Pr (>Chi)) for each predictor. Variables with a significant association (*p* < 0.05) are indicated with asterisks, with higher significance levels marked accordingly (*** for *p* < 0.001, ** for *p* < 0.01, for *p* < 0.1, and no asterisk for *p* ≥ 0.1.). X6: Whether a household member works in a human or veterinary health facility. X25: The type of diet the dog receives. X28: The inclusion of raw meat or bones in the dog’s diet. X35: Contact with animals not under the owner’s supervision. X38: Whether the dog is walked on a leash.

Variable	Df	Deviance	AIC	LRT	Pr (>Chi)
X6	2	196.62	208.62	30.52	<2 × 10^−16^ ***
X25	2	179.95	191.95	13.85	0.0010 ***
X28	1	173.91	187.91	7.81	0.0052 **
X35	1	173.29	187.29	7.19	0.0073 **
X38	1	175.75	189.75	9.65	0.0019 **

**Table 4 animals-15-01357-t004:** The results of the multivariable logistic regression analysis assessing risk factors associated with the presence of AMC-resistant *E. coli* in household dogs. The table presents the estimated regression coefficients, ORs, and their corresponding 95% confidence intervals (IC_2.5–IC_97.5). The *p*-values (Pr (>|z|)) indicate the statistical significance of each predictor in the model, with significance levels denoted as follows: ** for *p* < 0.01, * for *p* < 0.05, for *p* < 0.1, and no asterisk for *p* ≥ 0.1. X6: Whether a household member works in a human or veterinary health facility. X25: The type of diet the dog receives. X28: The inclusion of raw meat or bones in the dog’s diet. X35: Contact with animals not under the owner’s supervision. X38: Whether the dog is walked on a leash.

Variable	Coefficients	OR	IC_2.5	IC_97.5	Pr (>|z|)	Significance
(Intercept)	1.2358	3.441	0.772	15.303	0.099	.
X6Yes	−1.1512	0.316	0.129	0.717	0.008	**
X25Commercial food	−0.8765	0.416	0.123	1.526	0.164	
X25Homemade food	19.0878	194,866,108.8	2.92 × 10^−17^	1.01 × 10^249^	0.990	
X28Yes	−2.2851	0.102	0.005	0.562	0.034	*
X35Yes	−1.1322	0.322	0.129	0.744	0.011	*
X38Yes	−1.3533	0.258	0.109	0.604	0.002	**

**Table 5 animals-15-01357-t005:** The results of the likelihood ratio test (LRT) for variables included in the multivariable logistic regression model for the CTZ-resistant *E. coli* in household dogs. The table presents the degrees of freedom (Df), deviance, Akaike Information Criterion (AIC), LRT values, and *p*-values (Pr (>Chi)) for each predictor. Variables with a significant association (*p* < 0.05) are indicated with asterisks, with higher significance levels marked accordingly (** for *p* < 0.01, * for *p* < 0.05, for *p* < 0.1, and no asterisk for *p* ≥ 0.1). X6: Whether a household member works in a human or veterinary health facility. X10: The sex of the pet. X11: The age of the pet. X14: Whether the dog was purchased or adopted. X36: Whether the dog lives with other pets that have been hospitalized in the past year. X37: Whether the dog is taken for walks during the week.

Variable	Df	Deviance	AIC	LRT	Pr (>Chi)
X6	2	98.05	112.05	10.005	0.007 **
X10	1	92.05	108.05	4.009	0.045 *
X11	2	100.18	114.18	12.137	0.002 **
X14	1	97.24	113.24	9.201	0.002 **
X36	1	92.82	108.81	4.774	0.029 *
X37	1	93.41	109.41	5.370	0.021 *

**Table 6 animals-15-01357-t006:** The results of the multivariable logistic regression analysis assessing risk factors associated with the presence of CTZ-resistant *E. coli* in household dogs. The table presents the estimated regression coefficients, ORs, and their corresponding 95% confidence intervals (IC_2.5–IC_97.5). The *p*-values (Pr (>|z|)) indicate the statistical significance of each predictor in the model, with significance levels denoted as follows: ** for *p* < 0.01, * for *p* < 0.05, for *p* < 0.1, and no asterisk for *p* ≥ 0.1. X6: Whether a household member works in a human or veterinary health facility. X10: The sex of the pet. X11: The age of the pet. X14: Whether the dog was purchased or adopted. X36: Whether the dog lives with other pets that have been hospitalized in the past year. X37: Whether the dog is taken for walks during the week. NA: not available.

	Coefficient	OR	IC_2.5	IC_97.5	Pr (>|z|)	Significance
(Intercept)	−22.1098	2.50 × 10^−10^	0	4.16 × 10^52^	0.994	
X6Yes	0.1711	1.187	0.346	4.02	0.781	
X10Male	1.1230	3.074	1.024	10.29	0.053	.
X11Older than 10 years	−18.1551	1.30 × 10^−8^	NA	3.06	0.996	
X11Under 2 years old	1.3548	3.876	1.255	12.86	0.021	*
X14Purchase	1.7978	6.036	1.882	21.32	0.003	**
X36Yes	−18.7361	7.30 × 10^−9^	NA	2.60	0.996	
X37Yes	18.3003	88,661,041.94	3.28 × 10^−65^	2.81	0.995	

**Table 7 animals-15-01357-t007:** The results of the likelihood ratio test (LRT) for variables included in the multivariable logistic regression model for the ENR-resistant *E. coli* in household dogs. The table presents the degrees of freedom (Df), deviance, Akaike Information Criterion (AIC), LRT values, and *p*-values (Pr (>Chi)) for each predictor. Variables with a significant association (*p* < 0.05) are indicated with asterisks, with higher significance levels marked accordingly (*** for *p* < 0.001, ** for *p* < 0.01, * for *p* < 0.05, for *p* < 0.1, and no asterisk for *p* ≥ 0.1.). X6: Whether a household member works in a human or veterinary health facility. X12: The size of the pet. X15: The length of time the owner has been living with the pet. X17: Whether the pet has ever been hospitalized. X26: The primary location where the owner purchases pet food. X33: Whether the pet receives treats.

Variable	Df	Deviance	AIC	LRT	Pr (>Chi)
X6	2	249.79	267.79	62.666	<2 × 10^−16^ ***
X12	2	194.79	212.79	7.669	0.0216 *
X15	2	198.13	216.13	11.003	0.0041 **
X17	1	199.91	219.91	12.788	0.0003 ***
X26	2	197.51	215.51	10.384	0.0056 **
X33	1	191.76	211.76	4.637	0.0313 *

**Table 8 animals-15-01357-t008:** The results of the multivariable logistic regression analysis assessing risk factors associated with the presence of CTZ-resistant *E. coli* in household dogs. The table presents the estimated regression coefficients, ORs, and their corresponding 95% confidence intervals (IC_2.5–IC_97.5). The *p*-values (Pr (>|z|)) indicate the statistical significance of each predictor in the model, with significance levels denoted as follows: *** for *p* < 0.001, ** for *p* < 0.01, * for *p* < 0.05, for *p* < 0.1, and no asterisk for *p* ≥ 0.1. X6: Whether a household member works in a human or veterinary health facility. X12: The size of the pet. X15: The length of time the owner has been living with the pet. X17: Whether the pet has ever been hospitalized. X26: The primary location where the owner purchases pet food. X33: Whether the pet receives treats.

	Coefficient	OR	IC_2.5	IC_97.5	Pr (>|z|)	Significance
(Intercept)	0.46	1.58	0.25	9.91	0.621	
X6Yes	−1.78	0.17	0.07	0.37	0.000	***
X12Medium (between 11–27 kg)	−0.71	0.49	0.14	1.78	0.264	
X12Small (less than 11 kg)	0.44	1.55	0.50	5.34	0.464	
X15Between 5 and 10 years old	−1.30	0.27	0.11	0.65	0.005	**
X15More than 10 years	−1.31	0.27	0.07	0.88	0.041	*
X17Yes	1.45	4.24	1.91	9.88	0.001	***
X26Mass sales commercial establishment (supermarkets, pet-shop)	−2.00	0.14	0.03	0.50	0.004	**
X26Small-sized commercial establishment (neighborhood store, fair)	−1.04	0.36	0.06	1.90	0.235	
X33Yes	1.079	2.94	1.10	8.85	0.041	*

## Data Availability

The original contributions presented in this study are included in the article/Appendix A; further inquiries can be directed to the corresponding authors.

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
