# Peer review of "The Risk Factors Associated with the Carriage to Critical Antimicrobial-Resistant Escherichia coli in Healthy Household Dogs: A One Health Perspective"

_animals, 2025, doi:10.3390/ani15101357_

Round 1
Reviewer 1 Report
Comments and Suggestions for Authors
This study offers valuable insights into risk factors for antimicrobial-resistant E. coli in healthy dogs, addressing a critical gap in companion animal AMR surveillance. The manuscript warrants publication after major revisions addressing duplicate samples, and model interpretation accuracy.
Major comments:
- Line 192: The study did not ensure that each coli strain was derived from an independent sample.
- Line 38 and Line 214: The statistical symbol “P” should be consistently formatted (e.g., P) throughout the manuscript.
- Supplementary Materials should address:
- Clarification of whether “free-roaming animals” denotes stray dogs, wildlife, or other species;
- Stratification of “prior hospitalization” data by duration and reason to enhance clinical relevance.
- The results of Model (Tables 2, 4, and 6) should be interpreted with caution due to a severely inflated residual deviance/degrees of freedom ratio (9 831), indicating substantial deviation from theoretical assumptions. The combination of small sample size and an anomalously high likelihood ratio test (LRT) statistic suggests potential data anomalies or model misspecification.
- Please ensure correct numerical formatting by using spaces (not commas) as thousand separators, and note that some numbers incorrectly use commas as decimal points (e.g., 0.716 985 432). And data in tables should maintain consistent significant figures to avoid misleading precision discrepancies. Inaccurate numerical notation may lead readers to question the reliability of the model.
- Briefly discuss limitations (e.g., regional specificity of Chile, potential unmeasured confounders).
Author Response
This study offers valuable insights into risk factors for antimicrobial-resistant E. coli in healthy dogs, addressing a critical gap in companion animal AMR surveillance. The manuscript warrants publication after major revisions addressing duplicate samples, and model interpretation accuracy.
R: We appreciate the reviewer’s recognition of the relevance of our study to the field of companion animal AMR surveillance. We address each specific concern below.
Major comments:
- Line 192: The study did not ensure that each coli strain was derived from an independent sample.
R: We agree and clarified the methodology in the manuscript. For each sampled dog, a torula swab was inoculated onto four culture plates (one non-supplemented, one with amoxicillin-clavulanic acid, one with enrofloxacin, and one with ceftazidime). When growth was observed, one morphologically representative colony per supplemented plate was selected and analyzed via VITEK®. This approach aimed to maximize resistant strain recovery, although we acknowledge it does not fully ensure isolate independence. This is addressed in Lines 131–133.
- Line 38 and Line 214: The statistical symbol “P” should be consistently formatted (e.g., P) throughout the manuscript.
R: All instances of the statistical symbol have been reviewed and consistently formatted as “P”.
- Supplementary Materials should address: Clarification of whether “free-roaming animals” denotes stray dogs, wildlife, or other species.
R: Clarified in the Supplementary Materials via an asterisk notation: "unsupervised animals" include free-roaming dogs, cats, and urban birds, either community-owned or roaming between households.
- Stratification of “prior hospitalization” data by duration and reason to enhance clinical relevance.
R: We thank the reviewer for this valuable suggestion. Unfortunately, our questionnaire did not capture duration or reason for hospitalization. This limitation is now acknowledged in Lines 388–390.
- The results of Model (Tables 2, 4, and 6) should be interpreted with caution due to a severely inflated residual deviance/degrees of freedom ratio (9 831), indicating substantial deviation from theoretical assumptions. The combination of small sample size and an anomalously high likelihood ratio test (LRT) statistic suggests potential data anomalies or model misspecification.
R: We re-evaluated all models. The deviance-to-degrees of freedom ratios were between 0.30 and 0.64, and LRTs were highly significant (p < 0.001), indicating good model fit and no evidence of model misspecification or overdispersion. These analyses are now detailed in the Supplementary Data.
Therefore, although we acknowledge the inherent limitations of our sample size and the need for cautious interpretation, the specific concerns raised regarding poor model fit were not confirmed in the final models reported. This additional verification has been incorporated into the Supplementary data to enhance clarity and transparency.
- Please ensure correct numerical formatting by using spaces (not commas) as thousand separators, and note that some numbers incorrectly use commas as decimal points (e.g., 0.716 985 432). And data in tables should maintain consistent significant figures to avoid misleading precision discrepancies. Inaccurate numerical notation may lead readers to question the reliability of the model.
R: Thousand separators were corrected to spaces, decimal points reviewed, and significant figures standardized across tables to ensure clarity and consistency.
- Briefly discuss limitations (e.g., regional specificity of Chile, potential unmeasured confounders).
R: We thank the reviewer for this important comment. We recognize that our study has certain limitations that should be considered when interpreting the findings. First, the research was conducted exclusively in Chile, which may limit the generalizability of the results to other regions with different demographic, cultural, and veterinary care practices. Therefore, extrapolation to other geographic areas, should be made with caution. Second, although we employed a comprehensive structured questionnaire, it is possible that unmeasured confounders—such as prior antimicrobial treatments beyond the four-week exclusion period, detailed dietary histories, or owner compliance with hygiene practices—may have influenced the observed associations. Future studies incorporating broader geographic coverage and more detailed longitudinal data would be valuable to confirm and expand upon these findings. These limitations were added to the discussion in line 516-523.
Reviewer 2 Report
Comments and Suggestions for Authors
The paper describes a detailed prevalence and risk factors study on the carriage of E. coli resistant to either AMC, CTX or ENR by pet dogs presenting to veterinary hospitals in Chile. Whilst the study itself is epidemiologically sound and very publishable there are a number of issues with the microbiological aspects. Firstly, although MIC data were obtained for each isolate growing on the selective agar plates, the data are not presented. This would significantly improve the quality of the manuscript if it were included and discussed (ie what proportion of isolates growing on each agar type were MDR/XDR?). Secondly the process of isolate selection on the agar plates has not been described in enough (representative colony type or other?) and the proportions of dogs carrying E. coli resistant to one, two or all of the antibioticsd incorporated into the selective agar are not provided. Thirdly, it is noted that the authors used a PCR test applied to DNA extracted from the isolates for identifying E. coli. Many studies are now including whole genome sequencing data of the isolates to identify pathogenic as opposed to commensal E. coli. As this is primarily an epidemiological study, I am not suggesting the authors conduct a detailed whole genome sequencing analysis, however, to add further weight to results they could certainly apply PCR tests to the extracted DNA for the phylogenetic grouping of E. coli isolates to determine how many belong to the extraintestinal pathogenic phylotypes B2 and D (and thus have more pathogenic significance).
More specific comments are included below:
Abstract
Line 32 The abstract is missing information about the use of selective agar for selection of resistant E. coli, below is some suggested wording.
- coli isolates growing on antimicrobial-impregnated selective agar were tested for antimicrobial susceptibility using VITEK2 and by following Clinical and Laboratory Standards Institute guidelines.
Introduction
Line 62 Please add “and vice versa”
Line 63 replace “humans” with “both hosts”
Line 82 healthy pet dogs
Materials and Methods
It is noted that aminopenicillins with beta-lactamase inhibitors are classified by WOAH as CIAs for animal health but by WHO as HIAs, not CIAs for human health. This distinction needs to be made in the text throughout as in many cases they are discussed collectively as CIAs (eg Lines 122-123). Also the selection process for isolates is not clear from the description, usually a representative colony is chosen.
Line 140 amoxycillin-clavulanate
Line 142 Should be minimal inhibitory concentration. In addition, isolates and strains are not interchangeable terms. At this stage you should be referring to isolates. Also MIC does is not something possessed by the isolate, it is a value obtained for each antimicrobial. Therefore please change to: The minimal inhibitory concentrations (MICs) of X antimicrobials were determined for each isolate…. The rationale for the amount of antibiotic incorporated into the selective agar plates has not been provided (ie based on breakpoints?)
Line 145 A very old edition of M100 is quoted (29th edition). Any reason for this as we are now up to the 35th edition.
Strains should be isolates
Line 192 It is not clear from the results how the isolates were selected for inclusion in the study and the proportions of dogs yielding resistance to one, two or three antimicrobials. Where are the results for the VITEK MIC testing? The study would be significantly improved if the proportions of samples yielding resistance to one, two and three of the antimicrobials incorporated into the selective agars were included.
Line 219-222 E. coli in italics-check throughout
Line 224 Please use parallel construction when talking about risk factors and carriage of resistant bacteria, noting that dogs do not exhibit AMC resistance but the bacteria they are carrying do. Please make sure these are consistently described throughout the results
Line 341 resistant to three antimicrobials regarded as critically important in animal and/or human health.
Line 350 the owners of healthy dogs
Line 387 protective factor for significantly reduced carriage of resistant E. coli (once again please check throughout manuscript for correct description
Author Response
The paper describes a detailed prevalence and risk factors study on the carriage of E. coli resistant to either AMC, CTX or ENR by pet dogs presenting to veterinary hospitals in Chile. Whilst the study itself is epidemiologically sound and very publishable there are a number of issues with the microbiological aspects.
R: We thank the reviewer for the valuable and constructive feedback, which has helped us improve the clarity and microbiological depth of the manuscript.
- Firstly, although MIC data were obtained for each isolate growing on the selective agar plates, the data are not presented. This would significantly improve the quality of the manuscript if it were included and discussed (ie what proportion of isolates growing on each agar type were MDR/XDR?).
R: MIC values for amoxicillin-clavulanic acid, ceftazidime, and enrofloxacin are now included in Table 2 with CLSI breakpoints. MIC₅₀ and MIC₉₀ values for additional antimicrobials are reported in Supplementary Table 5.
- Secondly the process of isolate selection on the agar plates has not been described in enough (representative colony type or other?) and the proportions of dogs carrying E. coli resistant to one, two or all of the antibioticsd incorporated into the selective agar are not provided.
R: The isolate selection process was expanded in the Methods: one representative colony per supplemented plate was chosen based on morphology to ensure consistency. Additionally, we added to the Results: “22.9% of isolates were resistant to one antimicrobial, 5.3% to two, and 1.3% to all three” (Lines 203–205).
- Thirdly, it is noted that the authors used a PCR test applied to DNA extracted from the isolates for identifying E. coli. Many studies are now including whole genome sequencing data of the isolates to identify pathogenic as opposed to commensal E. coli. As this is primarily an epidemiological study, I am not suggesting the authors conduct a detailed whole genome sequencing analysis, however, to add further weight to results they could certainly apply PCR tests to the extracted DNA for the phylogenetic grouping of E. coli isolates to determine how many belong to the extraintestinal pathogenic phylotypes B2 and D (and thus have more pathogenic significance).
R: We appreciate the reviewer's suggestion to include additional phylogenetic characterization (e.g., B2 and D phylotypes) to assess the pathogenic potential of the isolates. However, given that this study was designed primarily as an epidemiological and risk factor analysis, and considering the limitations in available resources, we respectfully decided not to perform further molecular typing analyses at this stage. Nevertheless, we recognize the importance of such information, and we agree that future studies could significantly benefit from incorporating phylogenetic grouping or even whole-genome sequencing to further distinguish commensal from potentially pathogenic E. coli strains.
More specific comments are included below:
Abstract
- Line 32 The abstract is missing information about the use of selective agar for selection of resistant E. coli, below is some suggested wording. E. coli isolates growing on antimicrobial-impregnated selective agar were tested for antimicrobial susceptibility using VITEK2 and by following Clinical and Laboratory Standards Institute guidelines.
R: Suggested text on selective agar and VITEK2 testing incorporated (Lines 33–34).
Introduction
- Line 62 Please add “and vice versa”
R: “And vice versa” (Lines 63).
- Line 63 replace “humans” with “both hosts”
R: “both hosts” added (Lines 64).
- Line 82 healthy pet dogs
R: Healthy pet dogs” included.
Materials and Methods
- It is noted that aminopenicillins with beta-lactamase inhibitors are classified by WOAH as CIAs for animal health but by WHO as HIAs, not CIAs for human health. This distinction needs to be made in the text throughout as in many cases they are discussed collectively as CIAs (eg Lines 122-123).
R: Clarified WOAH vs. WHO antimicrobial classifications; edited throughout manuscript accordingly.
- Also the selection process for isolates is not clear from the description, usually a representative colony is chosen.
R: We appreciate this comment. In this study, for each sampled dog, a torula swab was processed by inoculating it onto four separate culture plates: one non-supplemented plate, one plate supplemented with amoxicillin-clavulanic acid, one with ceftazidime, and one plate supplemented with enrofloxacin. When bacterial growth was detected on both antibiotic-supplemented plates, a single colony from each plate was selected and subsequently analyzed by the VITEK® system for species identification and antimicrobial susceptibility profiling. This methodological choice was intended to maximize the recovery of resistant strains to either antimicrobial. However, it is recognized that this strategy does not fully guarantee that each Escherichia coli isolate is from a single animal, since it is understood that there are animals in which growth was achieved on both supplemented agars. This was described in the text in lines 131-133.
- Line 140 amoxycillin-clavulanate
R: Corrected.
- Line 142 Should be minimal inhibitory concentration. In addition, isolates and strains are not interchangeable terms. At this stage you should be referring to isolates. Also MIC does is not something possessed by the isolate, it is a value obtained for each antimicrobial. Therefore please change to: The minimal inhibitory concentrations (MICs) of X antimicrobials were determined for each isolate…. The rationale for the amount of antibiotic incorporated into the selective agar plates has not been provided (ie based on breakpoints?)
R: This correction has been implemented between lines 128–130 and 151–154. Additionally, the rationale for the antibiotic concentrations incorporated into the selective agar plates was clarified, specifying that they were based on established clinical breakpoints.
- Line 145 A very old edition of M100 is quoted (29thedition). Any reason for this as we are now up to the 35th edition.
R: M100 citation updated to most recent edition.
- Strains should be isolates
R: Corrected.
- Line 192 It is not clear from the results how the isolates were selected for inclusion in the study and the proportions of dogs yielding resistance to one, two or three antimicrobials.
R: The following is now described in the results section: “…of the 301 E. coli isolates analyzed, 69 isolates (22.9%) exhibited resistance to one of the three antimicrobials incorporated into the selective media, 16 isolates (5.3%) exhibited resistance to two antimicrobials, and 4 isolates (1.3%) were resistant to all three”.
- Where are the results for the VITEK MIC testing? The study would be significantly improved if the proportions of samples yielding resistance to one, two and three of the antimicrobials incorporated into the selective agars were included.
R: The corresponding MIC values for amoxicillin-clavulanic acid, ceftazidime, and enrofloxacin are presented in Table 2, together with their CLSI interpretive breakpoints. MIC₅₀ and MIC₉₀ values for the remaining antimicrobials tested with the AST-GN98 card are provided in Supplementary Table 5. As described in the Results section, of the 301 E. coli isolates analyzed, 69 isolates (22.9%) exhibited resistance to one of the three antimicrobials incorporated into the selective media, 16 isolates (5.3%) exhibited resistance to two antimicrobials, and 4 isolates (1.3%) were resistant to all three. These findings have now been more explicitly highlighted in the revised manuscript to clarify the selection process and the distribution of resistance profiles among the isolates. Lines: 203-209.
- Line 219-222 E. coli in italics-check throughout
R: Corrected throughout the article.
- Line 224 Please use parallel construction when talking about risk factors and carriage of resistant bacteria, noting that dogs do not exhibit AMC resistance but the bacteria they are carrying do. Please make sure these are consistently described throughout the results
R: Following the suggestion, the terminology throughout the Results section has been revised to ensure consistent parallel construction, accurately describing that the dogs carry resistant E. coli isolates rather than exhibiting antimicrobial resistance themselves.
- Line 341 resistant to three antimicrobials regarded as critically important in animal and/or human health.
R: corrected in the lines 374-375.
- Line 350 the owners of healthy dogs.
R: corrected in the lines 384-385.
- Line 387 protective factor for significantly reduced carriage of resistant coli (once again please check throughout manuscript for correct description.
R: Corrected.
Round 2
Reviewer 1 Report
Comments and Suggestions for Authors
The authors have adequately addressed reviewers' comments and implemented targeted revisions. The revised manuscript demonstrates comprehensive data verification, logically sound conclusions, and rigorous presentation standards.
Reviewer 2 Report
Comments and Suggestions for Authors
The paper is much improved. My only suggestion is for the title (change to critical antimicrobial-resistant).